# Exploring Effects of Chitosan Oligosaccharides on the DSS-Induced Intestinal Barrier Impairment In Vitro and In Vivo

**DOI:** 10.3390/molecules26082199

**Published:** 2021-04-11

**Authors:** Yujie Wang, Rong Wen, Dongdong Liu, Chen Zhang, Zhuo A. Wang, Yuguang Du

**Affiliations:** 1State Key Laboratory of Biochemical Engineering, Institute of Process Engineering, Chinese Academy of Sciences, Beijing 100190, China; wangyujie19@ipe.ac.cn (Y.W.); wenrong20210322@163.com (R.W.); harryldd@ipe.ac.cn (D.L.); 2School of Chemical Engineering, University of Chinese Academy of Sciences, Beijing 100049, China; 3College of Life Science, Sichuan Normal University, Chengdu 610101, China

**Keywords:** chitosan oligosaccharide, intestinal barrier, mucus, intestinal tight junction, inflammatory bowel disease

## Abstract

Intestinal barrier dysfunction is an essential pathological change in inflammatory bowel disease (IBD). The mucus layer and the intestinal epithelial tight junction act together to maintain barrier integrity. Studies showed that chitosan oligosaccharide (COS) had a positive effect on gut health, effectively protecting the intestinal barrier in IBD. However, these studies usually focused on its impact on the intestinal epithelial tight junction. The influence of COS on the intestinal mucus layer is still poorly understood. In this study, we explored the effect of COS on intestinal mucus in vitro using human colonic mucus-secreted HT-29 cells. COS relieved DSS (dextran sulfate sodium)-induced mucus defects. Additionally, the structural characteristics of COS greatly influenced this activity. Finally, we evaluated the protective effect of COS on intestinal barrier function in mice with DSS-induced colitis. The results indicated that COS could manipulate intestinal mucus production, which likely contributed to its intestinal protective effects.

## 1. Introduction

Inflammatory bowel disease (IBD) is becoming a public health challenge worldwide with its increasing incidence and prevalence [1,2]. About five million people in the world are diagnosed with IBD, which means that the effect of IBD on social health-care systems is significant, with life-long medication and poor quality of life of patients [3]. The pathogenesis of IBD is complex and still unclear, which is probably driven by intestinal barrier dysfunction, immune system disorders, and aberrant cross-talk between host and gut microbiota [4]. Several IBD patients display severe intestinal barrier dysfunction followed by enhanced antigen and bacterial invasion [5]. Many genetic studies also suggest the involvement of the gut barrier, as several susceptibility genes for IBD have key functions in the intestinal barrier and homeostasis, indicating that an impaired intestinal barrier has become a hallmark of IBD [6,7].

The intestinal barrier is mainly composed of two elements: the mucus layer and the intestinal epithelial tight junction [8]. The mucosal barrier is the first line preventing pathogen invasion in the gastrointestinal tract, consisting of gel-forming and transmembrane mucins [8]. MUC2 is the primary type of mucin secreted by the goblet cells in the colon. Several studies have found that mice lacking MUC2 displayed severe impaired intestinal barrier and colitis [9,10]. The intestinal epithelium is composed of monolayer cells and acts as a physical barrier against harmful antigens permeation that aggravates intestinal inflammation [11]. The defection of the intestinal epithelium, such as a lack of the tight junction protein occludin, will increase intestinal permeability and bacterial translocation, followed by gut inflammation [12]. Hence, it has become another pathogenic factor of IBD [7].

Several studies have shown that prebiotic interventions alleviated intestinal inflammation by modifying microbiota and modulating intestinal barrier function, promoting gut health. Prebiotics are fermentable dietary fibers, including both oligosaccharides and polysaccharides. In human studies, some researchers reported that a high-fiber diet could influence IBD outcomes [13,14]. Lindsay et al. found that fructooligosaccharide (FOS) supplementation influenced mucosal dendritic cell function [15]. Another study indicated that a purified glucomannan oligosaccharide from Amorphophallus konjac improved colonic mucosal barrier function by increasing the expression of the MUC2 gene [16]. Chitosan oligosaccharide (COS), the oligomers of β-(1-4)-linked D-glucosamine, is the degraded product of chitosan [17,18]. COS has attracted much attention and is widely used in the nutrition field due to its multifunctionality, high water solubility, non-toxicity, and biodegradability [19,20]. Moreover, many studies have highlighted that COS has a positive effect on gut health. Yousef et al. found that COS alleviated inflammation and associated intestinal barrier damage in both dextran sulfate sodium (DSS)-induced colitis and acetic acid-induced colitis animal models [21]. Interestingly, another study indicated that COS could promote the tight junction assembly of T84 cells [22]. Indeed, COS can also promote the integrity of the intestinal barrier in other models. Yang et al. showed that dietary supplementation of COS improved gut barrier function in the cecum of weaning pigs [23]. Our previous study showed that COS treatment recovered the decreased expression of occludin in the proximal colon of diabetic mice [24]. All of these studies suggest that COS has a beneficial effect on intestinal barrier function. However, these studies mainly focus on the outcome of COS treatment on the integrity of the epithelium barrier, whereas the impact of COS on the mucosal barrier remains obscure.

Here, inspired by DSS-induced disruption of the intestinal barrier in in vitro models [25,26] and in vivo models [21,27,28], we studied COS activity on DSS-induced mucus defects using human colonic mucus-secreted cells HT-29, to investigate whether COS alleviated intestinal barrier impairment by modulating intestinal mucus. Then, we compared the intestinal barrier restoration properties of COS with different molecular weights and degrees of deacetylation, respectively. Finally, we evaluated the protective effect of COS on the impaired intestinal barrier in IBD mice models.

## 2. Results

### 2.1. Effect of COS on the Expression of MUC2 and Occludin in HT-29 and Caco-2 Cells

We introduced 2% (wt/vol) of DSS to HT-29 and Caco-2 cells to recreate DSS-induced intestinal mucus and epithelium tight junction disruption and then investigated the effect of COS (chitosan oligosaccharides) on the expression of MUC2 and occludin in cells. As shown in Figure 1, DSS treatment significantly decreased the number of MUC2-positive HT-29 cells and occludin-positive Caco-2 cells. After incubation with 200 μg/mL COS for 24 h, we found that COS significantly alleviated DSS-induced damage on mucus by promoting MUC2 expression of HT-29 cells and increased the occludin expression of Caco-2 cells.

### 2.2. Comparison of Effect of COS with Dissimilar Structures on the Expression of MUC2 and Occludin in HT-29 and Caco-2 Cells

Several studies have shown the chemical characteristics of COS such as molecular weight and degree of deacetylation exert a significant influence on these biological properties [29,30,31]. Thus, we investigated the effect of 200 µg/mL of COS (low-molecular-weight chitosan oligosaccharide, molecular weight range = 363–1329 Da, degree of deacetylation >95%), HWCOS (high-molecular-weight chitosan oligosaccharide, molecular weight range = 4000–6000 Da, degree of deacetylation >90%), and NACOS (chitin oligosaccharide, molecular weight range = 300–1700 Da, degree of deacetylation <10%) on the expression of MUC2 and occludin in DSS-damaged cells. Compared with COS, HWCOS had a slighter effect, while NACOS had no significant beneficial effect on DSS-damaged cells both in mucus production and the epithelium tight junction (Figure 2).

We also used quantitative real-time PCR (qRT-PCR) to measure the effect of COS, HWCOS, and NACOS on the mRNA levels of the MUC2 gene in HT-29 cells and the OCLN gene in Caco-2 cells (Figure 3). The MUC2 level in the DSS-treated group was reduced to 0.6-fold of the control group and was recovered by COS treatment. HWCOS and NACOS had no noticeable effect. For the OCLN gene, COS greatly up-regulated its level in the DSS group, though HWCOS had a slight promotion effect. Additionally, NACOS treatment showed no positive impact on OCLN gene expression in the DSS group. These findings suggested that COS with low molecular weight and a high degree of deacetylation may display optimal effect on HT-29 and Caco-2 cells.

### 2.3. Cell Viability after COS, HWCOS and NACOS Treatment in HT-29 and Caco-2 Cells

We treated cells with a series of dilutions of COS, HWCOS, and NACOS (10, 100, 200, 500, 1000 μg/mL) to evaluate their effect on cell viability. We determined that COS showed no significant toxic effect on HT-29 and Caco-2 cells (Figure 4). High concentrations (500, 1000 μg/mL) of NACOS slightly inhibited HT-29 and Caco-2 cell viability, and 500 and 1000 μg/mL of HWCOS showed cytotoxicity to Caco-2 cells. Therefore, the 200 μg/mL concentration we selected in the above experiments did not significantly promote or inhibit the cell viability.

### 2.4. Effect of COS on Clinical Symptoms in Mice with DSS-Induced Colitis

The protective efficacy of COS in the treatment of IBD was assessed in the DSS-induced acute colitis model. To evaluate the effect of COS on clinical symptoms, C57BL/6 mice were given 4% (wt/vol) of DSS in drinking water and were intragastrically administered with COS at a dose of 200 mg/kg/day for 21 days. As indicated in Figure 5, all mice died after treated with DSS alone for 10 days. We also found the DSS-treated mice showed less body weight and higher disease activity index (DAI) scores than the control group. COS treatment slightly improved the survival rate at day 10, and inhibited the body weight change. Notably, COS markedly decreased the DAI score in mice with DSS-induced colitis after 7 days.

To further investigate the effect of COS on the intestinal barrier, H&E (hematoxylin and eosin) staining was performed to assess the histopathological alteration of the colon. As depicted in Figure 6, there was no significant histological change between the control and the COS-treated group. However, severed submucosal edema and the infiltration of inflammatory cells were observed in the DSS group (Figure 6, black frames). As expected, submucosal edema and the infiltration of inflammatory cells were alleviated by COS administration, which indicated that COS was effective in preventing severe intestinal damage in this model.

Finally, immunofluorescent staining of the intestinal tissue was performed to assess the effect of COS on the intestinal mucus and epithelium tight of mice with DSS-induced colitis. As shown in Figure 7, the intestinal tissue of the DSS group showed a remarkable reduction of intestinal mucus production quantitated by the MUC2 antibody and wheat germ agglutinin (WGA) labeled with FITC. The reduction of mucus was, though not fully recovered, apparently relieved in mice after COS treatment. Similarly, the tight junction of tissue stained with occludin antibody was also strengthened in the DSS+COS group. These results suggested that COS treatment significantly relieved intestinal barrier impairment in mice with DSS-induced colitis.

## 3. Discussion

In this study, we determined the direct protective effects of COS on the intestinal barrier. We demonstrated COS restoration on DSS-induced impairment of human colonic epithelial cell line Caco-2 and mucus-secreted cells HT-29. We found that low molecular weight COS with a high degree of deacetylation was more effective in recovering the MUC2 and occludin expression of cells. Cell viability assay indicated that 200 μg/mL COS, HWCOS, and NACOS did not promote HT-29 and Caco-2 cells’ viability, nor did they display any significant toxic effect on them. Finally, we demonstrated that COS administration increased the survival rate, suppressed the body weight loss, decreased the DAI score, improved histopathological manifestations, and significantly restored the intestinal barrier of colitis in DSS-induced IBD mice models.

The pathogenic process of IBD is a complex cascade of host–microbe and intercellular cross-talks in which the intestinal mucosal and epithelium barrier function may be crucial [32]. COS is the degraded product of chitosan and has many biological activities, especially in anti-inflammation [33,34,35]. Some studies indicated the protective effect of COS on the intestinal barrier but neglected its potential impact on intestinal mucin secretion. We assessed the change of MUC2 production to investigate the direct effect of COS on mucus, and this was the first study exploring the specific target of COS on the MUC2 expression of HT-29 cells. Many studies used 2–5% (wt/vol) of DSS to induce disruption of the intestinal barrier in in vitro models [25,26] and in vivo models [21,27,28]. In the preliminary experiments, 2%, 3%, 4%, and 5% (wt/vol) of DSS were used to evaluate its effect on cells. We found that 2% (wt/vol) of DSS could significantly induce cell impairment. Therefore, 2% (wt/vol) of DSS was chosen to conduct the experiments. Our results indicated that COS alleviated the DSS-induced disruption of mucus by increasing MUC2 secretion and promoting the MUC2 gene mRNA expression level of HT-29 cells. The cell viability assay showed that 200 μg/mL COS did not show any promotion or inhibition of cell viability, suggesting that COS directly affected cells to increase the gene expression and production of MUC2 and occludin (Figure 2 and Figure 3) rather than promote the cell viability. However, COS with 10 μg/mL and 100 μg/mL concentration had no significant effect on the intestinal barrier (MUC2 and occludin production) in the preliminary experiments. Therefore, COS with 200 μg/mL concentration was selected for further evaluation. Indeed, COS is a linear polymer composed of β-(1–4)-linked N-acetyl-D-glucosamine and D-glucosamine, both of which are related to the epithelium glycosylation [36]. Therefore, we speculate that COS could reach the colon and affect intestinal glycosylation, further affecting the MUC2 secretion [36]. Of course, the relationship between COS, altered glycosylation, and MUC2 synthesis and stability needs to be further discussed in the future. We also found that COS promoted the integrity of the intestinal tight junction by increasing occludin secretion and the OCLN gene mRNA level of Caco-2 cells, which was consistent with other previous studies [21,22]. In DSS-induced IBD mice models, drugs could be administrated before, during, or after the inducing of colitis to evaluate its prophylactic, protective, or therapeutic effects, respectively [21,28]. Furthermore, we evaluated the protective effect of COS on DSS-induced colitis in mice. In our study, the DSS group displayed a high mortality rate, which was similar to other studies [21,37,38]. Although COS did not significantly improve the survival rate or inhibit the weight loss of DSS-mice, it successfully decreased the DAI score and alleviated the submucosal edema and infiltration of inflammatory cells as described in previous studies [21,39,40]. Of note, the immunofluorescence staining of intestinal tissue suggested that COS could contribute to the prevention of severe intestinal barrier damage in DSS-mice. All these results showed the prospects for therapeutic efficacy of COS on intestinal barrier impairment. The relationship between its molecular weight, degree of deacetylation, and its intestinal barrier restoration properties also remain obscure. We compared the restoration properties of COS, HWCOS, and NACOS on the expression of MUC2 and occludin in HT-29 and Caco-2 cells. However, different from the significant effect of COS, HWCOS displayed a slight positive effect, and NACOS had no obvious change on either the protein levels or the mRNA levels of MUC2 and occludin. Indeed, many studies suggested that the degree of deacetylation and the molecular weight of COS exerted a notable influence on their biological activities [29,30]. Both Kidibule et al. [41] and Lee et al. [42] demonstrated that COS with 50% N-deacetylation displayed a lower anti-inflammatory effect than COS with 90% N-deacetylation. Pangestuti et al. [43] and Fernandes et al. [44] studied the relationship between the molecular weight of COS and its anti-inflammatory activity, concluding that low molecular weight COS was the most efficient. Our results are consistent with those studies mentioned above.

Of note, many animal studies highlighted that COS could protect the intestinal barrier by modulating gut microbiota according to their analysis of the gut microbiome [23,45], suggesting that COS could also have an indirect effect on intestinal mucus. Indeed, our previous study also found that COS markedly increased the abundance of *Akkermansia* in diabetic mice [24] and high-fat diet mice [46]. *Akkermansia* has been proven to degrade mucin, and the utilization of intestinal mucus stimulates more secretion of mucin by goblet cells, further enhancing the gut mucus barrier [47]. Hence, more studies need to focus on the abundance of mucus-related gut microbiota in IBD mice models, such as *Akkermansia* after treatment with COS, which better elucidates the indirect effect of COS on the intestinal barrier.

## 4. Materials and Methods

### 4.1. Preparation of COS, HWCOS and NACOS

COS was purchased from GlycoBio (GlycoBio, Dalian, China). NACOS was prepared in our lab in a previous study [48]. HWCOS was purchased from Sigma (St. Louis, MO, USA). Table 1 lists the molecular weight range and deacetylation degree of COS, HWCOS, and NACOS.

### 4.2. Cell Culture

The human colonic epithelial cell lines Caco-2 and HT-29 were purchased from the American Type Culture Collection (ATCC; Manassas, VA, USA). Caco-2 cells were cultured in Dulbecco’s modified Eagle medium containing 4.5 g/L glucose medium (DMEM/HG, Gibco) supplemented with 10% heat-inactivated fetal bovine serum (FBS, Gibco), penicillin (100 units/mL, Gibco), and streptomycin (100 µg/mL, Gibco). HT-29 cells were cultured in DMEM/F12 medium (DMEM/F12, Gibco) containing 5% heat-inactivated fetal bovine serum, penicillin (100 units/mL, Gibco), and streptomycin (100 µg/mL, Gibco). Both cells were grown at 37 °C in an incubator supplemented with 5% CO_2_. To recreate the dextran sodium sulfate (DSS, Sigma-Aldrich)-induced cell mucosal and epithelial damage models, 2% (wt/vol) DSS was introduced into the cells for 24 h when the cells were grown to 90% confluence in a 24-well plate.

### 4.3. Co-Culture of COS, HWCOS, NACOS and Cells

The old cell culture medium was washed off with phosphate-buffered saline (PBS; pH = 7.4) after establishing the cell mucosal and epithelial damage models. For the COS, HWCOS, and NACOS treatment on cells experiment, 200 μg/mL COS, HWCOS, and NACOS were added into the cell culture medium, respectively, to coculture with the cells for 24 h.

### 4.4. Cell Viability Assay

Cells were seeded in 96-well plates at a density of 1 × 10^4^ cells/well. After incubation, the cells were treated with different concentrations of COS, HWCOS, and NACOS (0, 10, 100, 200, 500, 1000 µg/mL) for 24 h. Then, 0.5 mg/mL MTT (M5655, Sigma-Aldrich) was added into the medium and the cells were incubated for 4 h in a 37 °C incubator as previously described [49]. Dimethyl sulfoxide (DMSO, 150 µL/well) was used to dissolve the formazan crystal. The cell viability was measured at 570 nm by a microplate reader.

### 4.5. Immunofluorescence Staining

Immunofluorescent staining of cells was performed according to the pervious study with minor revision [49]. After cultivation, the cells were treated with 4% paraformaldehyde (28908, Thermo Fisher Scientific, Waltham, MA, USA) for 30 min, and then the cells were ruptured with 0.3% Triton-X-100 at room temperature for 5 min. Next, the cells were blocked with 5% BSA (30036727, Thermo Fisher Scientific, Waltham, MA, USA) for 1 h. To observe the change in the intestinal epithelial mucus layer, HT-29 cells were incubated with Mucin2 monoclonal antibody (sc-7314, Santa Cruz Biotechnology, Dallas, TX, USA) at a dilution of 1:50 for 2 h. Then, the cells were washed with PBS and incubated with Alexa FluorTM488 linked goat anti-mouse IgG (H + L) super clonal secondary antibody (A11001, Invitrogen) at a 1:100 dilution for 1 h. To visualize intestinal epithelial tight junctions, Caco-2 cells were incubated with occludin monoclonal antibody (sc-133256, Santa Cruz Biotechnology, Dallas, TX, USA) at a dilution of 1:50 for 2 h. Then, the cells were washed with PBS and incubated with Alexa FluorTM594 linked goat anti-mouse IgG (H + L) super clonal secondary antibody (A11005, Invitrogen) at a 1:100 dilution for 1 h. After that, the cells were washed with PBS 3 times, and then the nuclei were stained with DAPI (D1306, Invitrogen) for 10 min. Images were taken using a Leica TCS SP8 STED 3X laser scanning confocal immunofluorescence microscope. All these images represented the typical situation of three independent experiments. In one experiment, each sample had three parallel tests, and three random images were taken in each test. For the fluorescent analysis of the images, the average immunofluorescence value of all images from one sample was calculated. Computerized quantification of the mean fluorescence intensity and the coverage rate of images was analyzed using Image J.

Immunofluorescent staining of intestinal tissue was performed according to the pervious study with minor revision [50]. The intestinal tissue sections were dewaxed and hydrated with xylene and different concentrations of ethanol, followed by the removal of endogenous peroxidase in 0.75% H_2_O_2_. The slices were then placed into boiling citric acid solution (0.01 M) for 30 min for antigen recovery. The sections were ruptured with 0.3% Triton-X-100 at room temperature for 30 min and then blocked with 10% goat serum for 30 min. To observe the change of the intestinal epithelial mucus layer, the tissue slides were incubated with Mucin2 monoclonal antibody (sc-7314, Santa Cruz Biotechnology, Dallas, TX, USA) at a dilution of 1:50 for 1 h. Then, the slides were washed with PBS and incubated with Alexa FluorTM488 linked goat anti-mouse IgG (H + L) super clonal secondary antibody (A11001, Invitrogen) at a 1:100 dilution for 30 min. For wheat germ agglutinin (WGA, L4895, Sigma-Aldrich, St. Louis, MO, USA) staining, the tissue slides were incubated with fluorescein isothiocyanate-wheat germ agglutinin (FITC-WGA) at a concentration of 10 μg/mL for 1 h [51]. To observe the change of the intestinal epithelial tight junction, the tissue slides were incubated with occludin monoclonal antibody (sc-133256, Santa Cruz Biotechnology, Dallas, TX, USA) at a dilution of 1:50 for 1 h. Then, the slides were washed with PBS and incubated with Alexa FluorTM488 linked goat anti-mouse IgG (H + L) super clonal secondary antibody (A11005, Invitrogen) at a 1:100 dilution for 30 min. Finally, the coverslips were stained with anti-fade reagent DAPI, and the images were acquired using a Leica DFC310 FX digital camera connected to a Leica DMI4000 B light microscope (Wetzlar, Germany). All these images represented the typical situation of three independent experiments. In one experiment, each sample had three parallel tests and three random images were taken in each test. For the fluorescent analysis of images, the immunofluorescence average value of all images from one sample was calculated. Computerized quantification of the mean fluorescence intensity and the coverage rate of images was analyzed using Image J.

### 4.6. RNA Extraction and Quantitative RT-PCR

Total RNA from intestinal epithelial cells was isolated according to the protocol of the Ultrapure RNA Kit (CoWin Biosciences, Beijing, China). The isolated RNA was reversely transcribed into cDNA using a HiFiScript cDNA Synthesis Kit (CoWin Biosciences, Beijing, China). The quantitative real-time PCR (qRT-PCR) analysis was performed by using a 7500 Fast Real-Time PCR System (Applied Biosystems, Foster City, CA, USA). The application for 40 cycles was as follows: 95 °C for 2 min, 95 °C for 15 s and 60 °C for 60 s. The sequences of the primers used were summarized in Table 2. Relative mRNA expressions were analyzed using the 2^−ΔΔCT^ method. All results were normalized to GAPDH [51].

### 4.7. Induction of Colitis and Treatment with COS

Male C57BL/6 mice (6–8 weeks, 20–22 g) were purchased from Beijing Huafukang Bioscience Co. Inc. and housed in a controlled environment with constant temperature (23 ± 2 °C) and humidity. After 2 weeks of adaptation, the mice were divided into four groups (*n* = 8). For the control group, the mice were treated with water for 21 days. For the DSS group, the mice were given 4% (wt/vol) of DSS in drinking water for 7 consecutive days and then treated with water for 14 days. For the COS group, the mice were intraperitoneally injected with COS at a dose of 200 mg/kg/day for 21 days. For the DSS + COS group, the mice were given 4% (wt/vol) of DSS in drinking water for 7 consecutive days and intraperitoneally injected with COS at a dose of 200 mg/kg/day for 21 days at the same time. The COS solution was changed every 2 days to maintain their bioactivities. The survival rate and body weight change of all mice were recorded during the treatment period.

### 4.8. Assessment of Severity of Colitis

Disease severity was assessed using the disease activity index (DAI) according to published methods [52]. For the hematoxylin and eosin (H&E) staining, the colon tissues were washed with PBS and then fixed in paraformaldehyde for 24 h and paraffin-embedded. Then, the colon specimens were cut into slices and stained with H&E.

### 4.9. Statistical Analysis

The Graphpad Prism 6 software (La Jolla, CA, USA) was used to analyze all data. Results were presented as mean ± SD. The significant differences between the two groups were assessed by the unpaired two-tailed Student’s *t*-test. A one-way ANOVA with post hoc Tukey’s test was used to measure the significant differences between the four compared groups. For all data, *p*-value < 0.05 was considered to be statistically significant.

## 5. Conclusions

In this study, COS with low molecular weight and a high degree of deacetylation displayed the biological activities with the best effect on the restoration of the intestinal barrier in vitro and in vivo, which suggests that the mitigation of COS on intestinal barrier damage can be a potential way to ameliorate colitis.

## Figures and Tables

**Figure 1 molecules-26-02199-f001:**
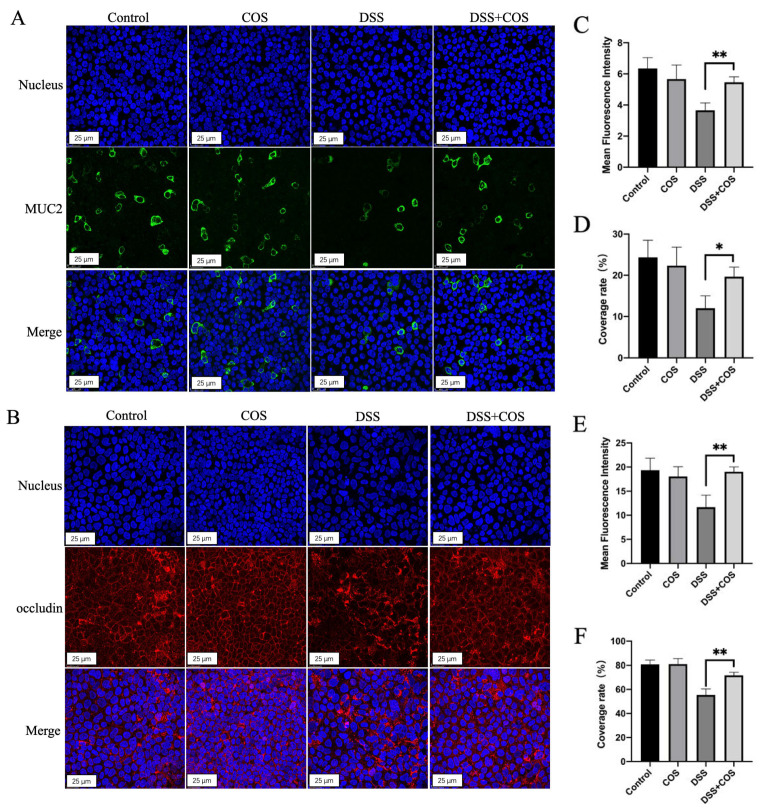
Effect of COS (chitosan oligosaccharide) on the expression of MUC2 and occludin in HT-29 and Caco-2 cells. (**A**) Immunofluorescence views of MUC2 (green) secreted by HT-29 cells in the control, COS, DSS (dextran sulfate sodium), and DSS + COS groups; (**B**) immunofluorescence views of occludin (red) secreted by Caco-2 cells in the control, COS, DSS, and DSS + COS groups; computerized quantification of the mean fluorescence intensity and the coverage rate of MUC2 (**C**,**D**) or occludin (**E**,**F**) was performed. The average value of randomly taken images was calculated (*n* = 9, * *p* < 0.05, ** *p* < 0.01). Error bar represents the standard deviation. Images represent the typical situation of three independent experiments. Scale bars, 25 µm.

**Figure 2 molecules-26-02199-f002:**
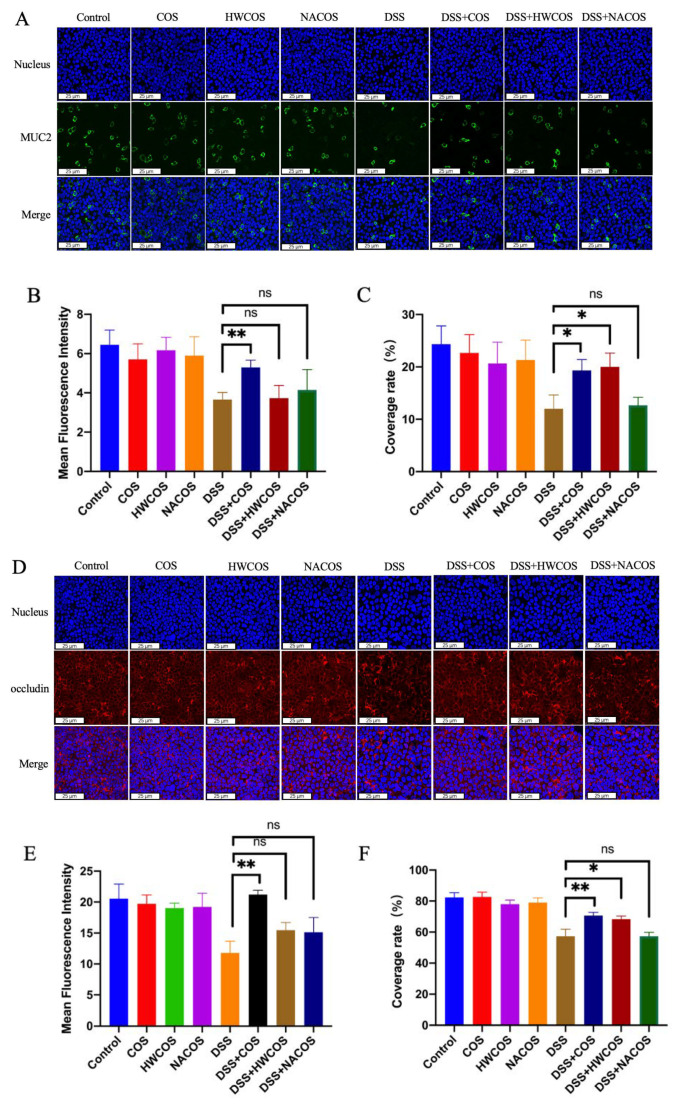
Effect of COS, HWCOS (high-molecular-weight chitosan oligosaccharide), and NACOS (chitin oligosaccharide) on the expression of MUC2 and occludin in HT-29 and Caco-2 cells. (**A**) Immunofluorescence views of MUC2 (green) secreted by HT-29 cells in the control, COS, HWCOS, NACOS, DSS, DSS + COS, DSS + HWCOS, and DSS + NACOS groups; (**D**) immunofluorescence views of occludin (red) secreted by Caco-2 cells in the control, COS, HWCOS, NACOS, DSS, DSS + COS, DSS + HWCOS, and DSS + NACOS groups; computerized quantification of the mean fluorescence intensity and the coverage rate of MUC2 (**B**,**C**) or occludin (**E**,**F**) was performed. The average value of randomly taken images were calculated (*n* = 9; * *p* < 0.05, ** *p* < 0.01). Error bar represents the standard deviation. Images represent the typical situation of three independent experiments. Scale bars, 25 μm.

**Figure 3 molecules-26-02199-f003:**
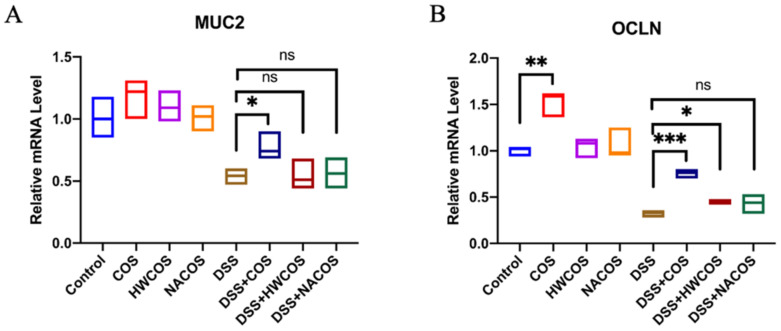
Effect of COS, HWCOS, and NACOS on the mRNA level of MUC2 in HT-29 and OCLN in Caco-2 cells. Cells were damaged by 2% (wt/vol) of DSS for 24 h, followed by treatment with 200 μg/mL of COS, HWCOS, and NACOS. After 24 h of incubation, the mRNA level of MUC2 in HT-29 cells (**A**) and OCLN in Caco-2 cells (**B**) were determined by quantitative real-time PCR. The average value of the data was calculated (*n* = 3; ns, not significant, * *p* < 0.05, ** *p* < 0.01, *** *p* < 0.001). Error bar represents the standard deviation. Data represent the typical situation of three independent experiments.

**Figure 4 molecules-26-02199-f004:**
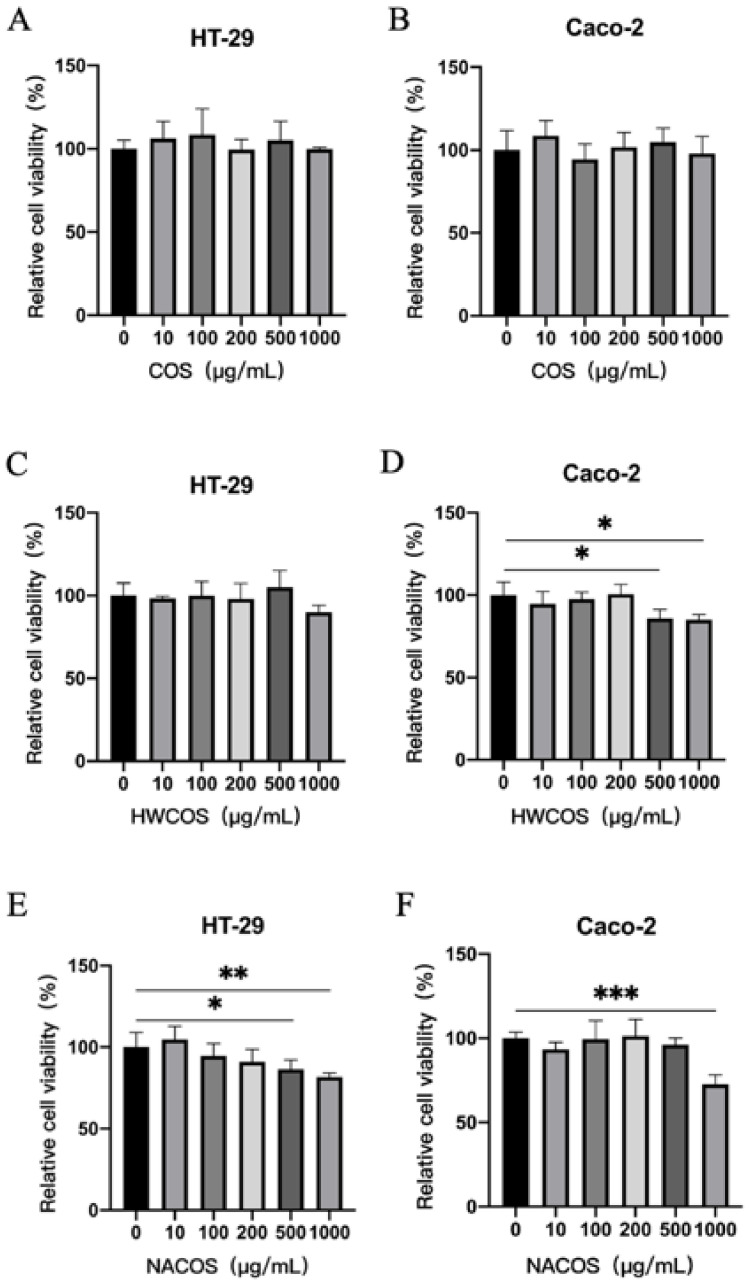
Cell viability after COS, HWCOS, and NACOS treatment in HT-29 and Caco-2 cells. Cells were treated with different concentrations (10, 100, 200, 500, 1000 μg/mL) of COS, HWCOS, and NACOS for 24 h. After 24 h of incubation, an 3-(4,5-dimethylthiazol-2-yl)-2,5-diphenyltetrazolium bromide (MTT) assay was performed to measure the HT-29 cell viability of COS (**A**), HWCOS (**C**), and NACOS (**E**), and the Caco-2 cell viability of COS (**B**), HWCOS (**D**), and NACOS (**F**). The average value of the data was calculated (*n* = 5; * *p* < 0.05, ** *p* < 0.01, *** *p* < 0.001). Error bar represents the standard deviation. Data represent the typical situation of three independent experiments.

**Figure 5 molecules-26-02199-f005:**
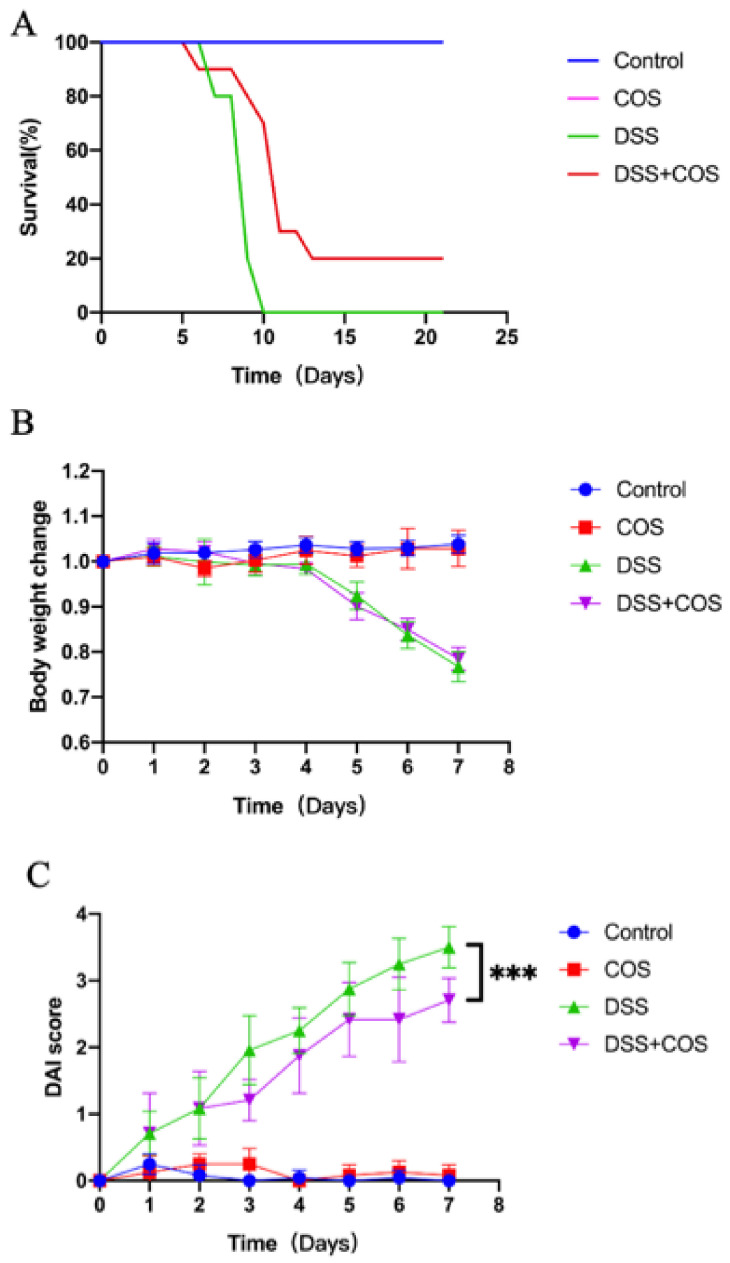
Treatment with COS improved clinical symptoms of DSS-induced colitis. Mice were given 4% (wt/vol) of DSS in drinking water for 7 consecutive days and intraperitoneally injected with COS at a dose of 200 mg/kg/day for 21 days at the same time. (**A**) Effect of treatment with COS on the survival rate of mice at 21 days. (**B**) Effect of treatment with COS on the body weight change of mice at 7 days. (**C**) Effect of treatment with COS on the DAI score of mice at 7 days. Results were presented as mean ± SD (*n* = 8; ns, not significant, *** *p* < 0.001).

**Figure 6 molecules-26-02199-f006:**
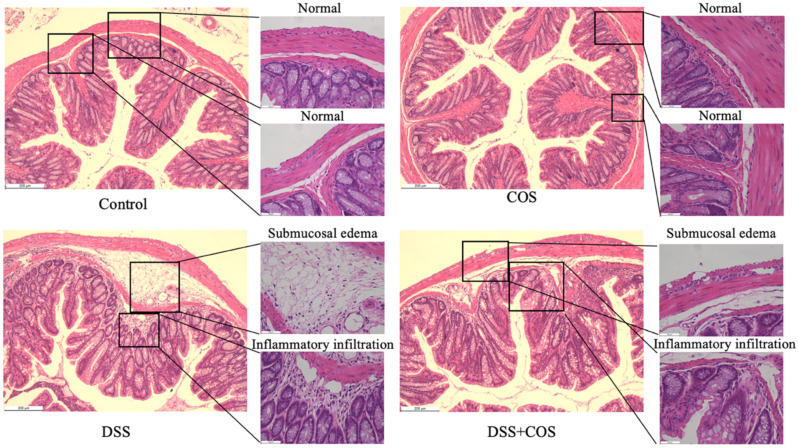
Effect of treatment with COS on histopathological alteration in DSS-induced colitis. Mice were given 4% (wt/vol) of DSS in drinking water for 7 consecutive days and intraperitoneally injected with COS at a dose of 200 mg/kg/day for 21 days at the same time. The histological changes of the colon in the control, COS, DSS, and DSS + COS groups were imaged by H&E (hematoxylin and eosin) staining. Black frames indicate normal structure, submucosal edema, and the infiltration of inflammatory cells. Scale bars, 200 μm and 50 μm.

**Figure 7 molecules-26-02199-f007:**
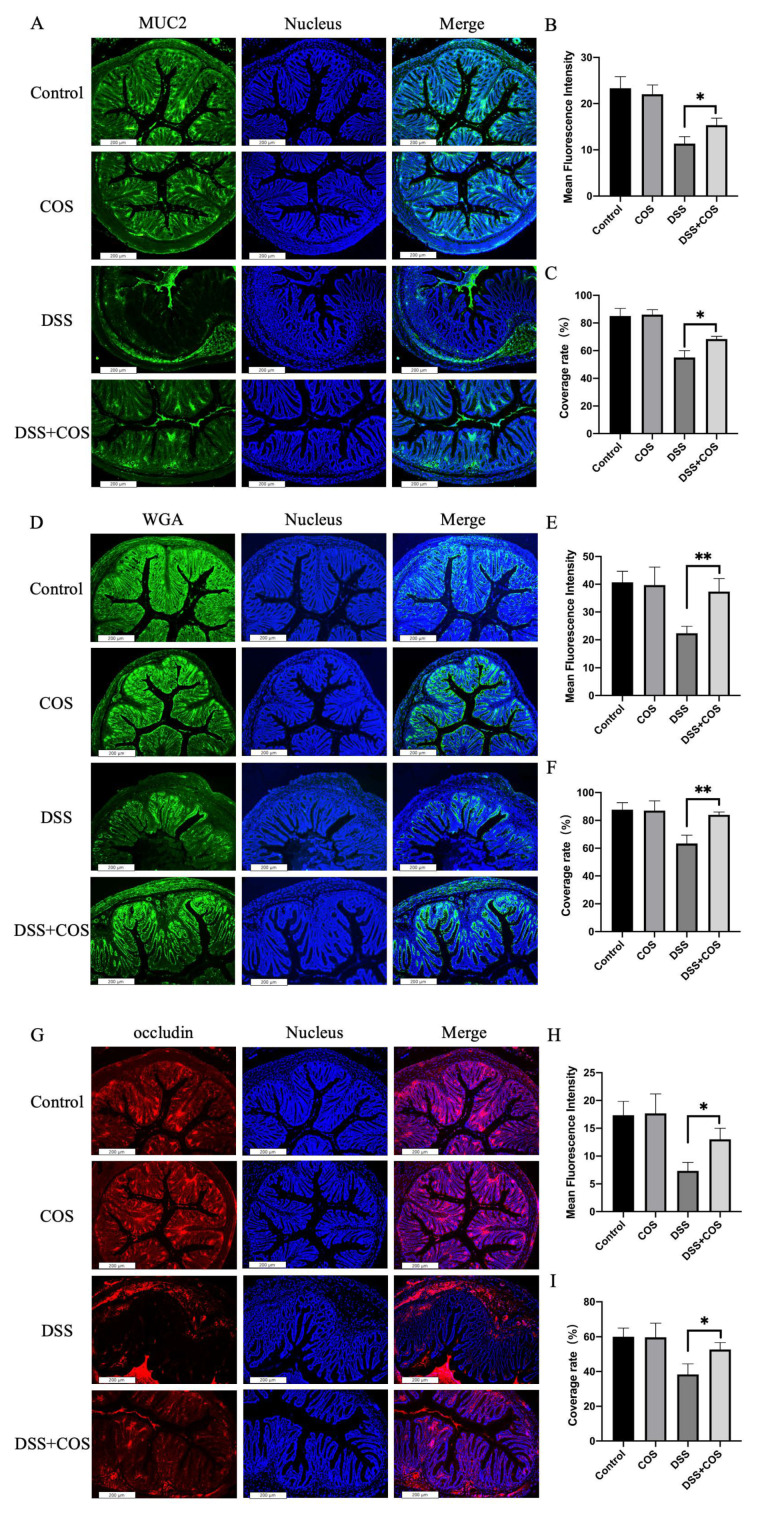
Treatment with COS restored the intestinal barrier damage of DSS-induced colitis. Immunofluorescence views of MUC2 (green) (**A**), FITC-WGA (green) (**D**), and occludin (red) (**G**) in intestinal tissue; computerized quantification of the mean fluorescence intensity and the coverage rate of MUC2 (**B**,**C**), FITC-WGA (**E**,**F**), and occludin (**H**,**I**) was performed. The average value of randomly taken images were calculated (*n* = 9; * *p* < 0.05, ** *p* < 0.01). Error bar represents the standard deviation. Images represent the typical situation of three independent experiments. Scale bars, 200 μm.

**Table 1 molecules-26-02199-t001:** The molecular weight range and deacetylation degree of COS, HWCOS, and NACOS.

Name	Molecular Weight Range	Deacetylation Degree
COS	363–1329 Da	>95%
HWCOS	4000–6000 Da	>90%
NACOS	300–1700 Da	<10%

**Table 2 molecules-26-02199-t002:** The oligonucleotide primer sequences used in qRT-PCR.

Gene	Sequences	
MUC2	F: GACCCGCACTATGTCACCTT	R: GGACAGGACACCTTGTCGTT
OCLN	F: ACAAGCGGTTTTATCCAGAGTC	R: GTCATCCACAGGCGAAGTTAA
GAPDH	F: GTGAAGGTCGGAGTCAACG	R: TGAGGTCAATGAAGGGGTC

## Data Availability

The data presented in this study are available on request from the corresponding author.

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
