# Peer review of "Exploring Effects of Chitosan Oligosaccharides on the DSS-Induced Intestinal Barrier Impairment In Vitro and In Vivo"

_molecules, 2021, doi:10.3390/molecules26082199_

Round 1

Reviewer 1 Report

In their study „Exploring Effects of Chitosan oligosaccharides on the DSS-induced intestinal barrier impairment in vitro and in vivo”, Wang et al. use two cell lines to investigate the effect of different Chitosan oligosaccharides (COS) on Muc2 production in HT-29 cells and occluding levels in Caco-2 cells. The authors also use an in vivo DSS-induced colitis model to test their in vitro generated results. The paper concludes that COS has a direct protective effect on the intestinal barrier in vitro and in vivo. COS is stated to restore DSS-induced impairments in cells and that COS effectiveness inversely correlated with its degree of acetylation. In mice, COS is stated to increase survival, decrease weight loss, and improve DAI in the DSS-induced colitis model.

The study is interesting and addresses an important disease-related topic. Protective roles of COS in DSS-induced models of colitis have previously been established, but the role of the cellular mucus layer has not been addressed in detail before. However, having said this, I have a number of concerns that need to be addressed to better support the claims of the authors:

Major:

Figure 1: Individual micrographs for the various cell lines and conditions are shown. However, it is unclear to the reader whether these micrographs represent the typical situation - quantification of several randomly chosen images is needed.

Do these pictures represent entire confocal stacks or individual layers?

Why do only few HT-29 cells produce Muc2 and the others do not? Should they not behave all in the same manner?

Figure 2: Quantification is shown here, but it is unclear how this quantification has been achieved, on how many images, and whether these images represented complete stacks.

Have biological or technical replicates been used for all quantifications?

In the section that described cell viability, the authors often use the term “growth” instead. Is this intended?

Using the mouse model and H&E stained intestinal epithelium, the authors report restored impairment of intestinal epithelium, improved submucosal edema, and less fibrosis and destruction of crypts in COS-treated DSS-mice. The presented figure 6, however, does not allow me to assess these improvements. All I can see from the in vivo study is that COS is not of sufficient activity to significantly improve induced colitis, although one mouse seemed to survive much longer that all others in the COS/DSS group (why is that?).

Therefore, the claimed findings, as again presented in the discussion, are overreaching. Especially the claims made for the in vivo effects need much better (direct) support, especially in terms of the functionality of the known players in epithelial barrier function. Immunohistological stainings including their quantification is needed.

Reviewer 2 Report

In this manuscript of "Exploring Effects of Chitosan Oligosaccharides on the DSS-induced intestinal barrier impairment in vitro and in vivo" authors explained that COS with low molecular weight and high degree of deacetylation displayed the best effect biological activities on the restoration of the intestinal barrier. A lot of experimental work was done and it is certainly contributing to the current scientific knowledgebase. But this needs revision and the author reply comments properly before accepted. The comments and questions are as follows:

  1. In 2.1, why choose 2% (wt/vol) of DSS to conduct experiment? Please describe more detail or add your references.
  2. In 2.3, the relative cell viability at 200 μg/mL concentration actually showed not significantly differences with 10 μg/mL and 100 μg/mL. May authors try to explain the proper reason for choosing this concentration?
  3. In cellular experiments, some results just showed not significant in Fig.2~4. Whether the insignificance of the results would affect the analysis of the experimental results and the subsequent in vivo experiments? Also, please check Figure 4 properly (typed as Figure 1).
  4. In Figure 5B, why were the indicator (body weight change) for the Control different from the other groups at the beginning (0 day)?
  5. In 4.4, the low concentration of MTT is rarely seen. Are there any references or protocols to support this concentration?
  6. In 4.7, authors choose given DSS while injected with COS simultaneously for DSS + COS mice group. Why were the animal modeling and drug administration steps performed simultaneously? Are there references to support this experimental approach? Please describe more detail about it.

Above all, I suggest this can be accepted before making revision or answer the questions properly.

Round 2

Reviewer 1 Report

The authors present a revised, strongly improved manuscript and have addressed most of my concerns. However, some minor concerns remain:

1) Line 182: ...infiltration of inflammatory cells were observed in the DSS group (Figure 6, arrows).

The authors have added arrows but I still cannot see anything particular in these figures. Can a blow-up of the areas of interest be provided?

2) Line 198: ...prevented in mice...

"prevention" seems overreaching to me, as there are still clearly visible tissue changes. The statement should be re-phrased to make clear that  no complete reversal or prevention is obseved in this system.

3) Line 259: ....repair...

It is unclear whether this is really repair or whether it rather is prevention of severe damage that is described in this study. Please correct.
